# Examination of the Effectiveness of the Healthy Empowered Active Lifestyles (HEAL) Program on Advanced Glycation End Products

**DOI:** 10.3390/ijerph18094863

**Published:** 2021-05-02

**Authors:** Mathew J. Gregoski, Janis Newton, Kathleen Blaylock, Sheila A. O. Smith, David P. Turner

**Affiliations:** 1Department of Public Health Sciences, College of Medicine, Medical University of South Carolina, Charleston, SC 29425, USA; 2Wellness Center, Medical University of South Carolina, Charleston, SC 29425, USA; newtonj@musc.edu (J.N.); katiemblaylock@gmail.com (K.B.); 3College of Nursing, Medical University of South Carolina, Charleston, SC 29425, USA; aquapreg@gmail.com; 4Department of Pathology and Laboratory Medicine, College of Medicine, Medical University of South Carolina, Charleston, SC 29425, USA

**Keywords:** lifestyle, physicalactivity, nutrition, health, food, advanced glycation end products

## Abstract

This pilot study investigated the effectiveness of the healthy empowered active lifestyles (HEAL) program to reduce circulatory levels of advanced glycation end products (AGEs) and assess its relationship to BMI, % body fat, fasting glucose, and A1C. The HEAL program was delivered at a local wellness center using a team-based approach and focused on physical activity and dietary education. A sample of twenty primarily European American (19 white, 1 black) participants (i.e., 10 males, 10 females) aged 26 to 71 (m = 48.75 ± 10.26) completed 12 weeks of the HEAL intervention. Pre to post changes in AGEs, BMI, % body fat, fasting glucose, and A1C were examined as primary outcomes. The findings showed participants had the following average reductions: AGEs 36.04 ± 18.48 ug/mL, BMI 2.0 ± 1.2 kg/m^2^, % body fat 3.18 ± 1.57%, fasting glucose 5.9 ± 17.21 mg/dL, and A1C 0.68 ± 1.11%. All twenty participants successfully completed the entire twelve weeks of the HEAL intervention. The results of this study show that the HEAL intervention provides beneficial reductions of AGEs, BMI, % body fat, fasting glucose, and A1C. In addition, the high adherence shows promise, and demonstrates the potential for HEAL as a behavioral intervention to improve pre-diabetic and other inflammatory related comorbidities. Further replication of results via additional randomized controlled trials is needed.

## 1. Introduction

Advanced glycation end products (AGEs) are reactive metabolites formed by the non-enzymatic glycosylation of biological macromolecules during normal metabolism and by increased levels of oxidative stress [1,2,3]. Cellular clearance of AGE metabolites is inefficient, and they accumulate in our plasma and tissue as we grow older. Increased AGE levels are associated with pro-inflammatory phenotypes and are linked with premature mortality, stroke, cardiovascular disease, diabetes, neurodegenerative disorders, and cancer [4,5,6,7,8,9,10]. Furthermore, dyslipidemia, high blood pressure, hyperglycemia, and obesity are critical components of metabolic syndrome that play a fundamental role in increasing AGE accumulation levels in our bodies [11].

Critically, AGE accumulation is also associated with lifestyle. The modern diet defined by high intake of highly processed, high fat, high sugar foods is high in AGE levels and represents a significant source of exogenous AGEs [12,13]. Additionally, frying, grilling, and broiling accelerate the formation of AGEs in food during cooking. Multiple studies have associated dietary-derived AGEs with the onset of and complications associated with diabetes, cardiovascular disease, and cancer risk [4,5,6,8]. Human and animal models indicate that increased physical activity may reduce the AGEs found in the circulatory system. In obese Zucker rats, regular moderate exercise reduced AGE levels and advanced glycation early diabetic nephropathy [14]. In non-diabetic middle-aged women, a 12-week lifestyle modification study showed that increases in the daily average number of steps taken correlates with lower AGE accumulation levels [15]. Meta-analysis studies demonstrate that weight loss in obese and overweight patients with type 2 diabetes is accompanied by a reduction in glycated hemoglobin (HbA1c) in a dose dependent manner, and is more effective in in populations with poor glycemic control than in well controlled populations [16]. In addition, glycation introduces both morphology and functional changes in adipose tissue, which are associated with metabolic and vascular alterations, as well as insulin resistance [16].

To date, little information is available regarding changes in AGEs as a function of participation in a physical activity and weight loss intervention that has continuously demonstrated successful outcomes across multiple cohorts of participants.

The current study examined the adaptation of a behavioral change intervention program that helps participants focus their diet and exercise behaviors to build a healthy empowered active lifestyle (HEAL) on AGE levels. HEAL was developed as part of a collaborative effort between MUSC researchers and the MUSC wellness center using Pender’s health promotion model, along with multiple behavioral change techniques to maximize changes towards leading a healthy lifestyle. Since the start in 2008, over 1000 participants have completed the program, losing over 30,000 pounds. A follow up study conducted on past participants reported that 70% participate in physical activity greater than 5 times per week, and 66% have maintained or continued to lose weight [17]. The program used to develop HEAL (i.e., the healthy Charleston challenge) has previously been recognized nationally as a successful community-based lifestyle intervention program [17].

Participants who enroll in HEAL are assigned to a team of 8–10 members that consists of a trainer, one to two mentors (past HEAL participants who have been successful), and other participants. The teams are created based on the workout time that each member is able to adhere to. Participants work out for one hour daily with their trainers, mentors, and other participants as a team for 12 weeks. Workouts are widely variable and include aerobic (e.g., running, stairs with weights) and anaerobic activities (e.g., group circuit training, weight bearing exercises, etc.). Participants are strongly encouraged to explore multiple physical activities in order to find activities they enjoy.

In addition to their fellow teammates, participants are also assigned a dietician to review their dietary intake and make adjustments throughout the program. Emphasis is placed on participants eliminating foods that are highly processed, high in fat and/or sugar, and nutritionally sparse. Once a week, all the teams meet together for educational sessions related to diet, exercise, disease prevention, and behavioral change techniques to maximize success, which are described in Table 1. During these weekly sessions, participants also complete weigh-ins and teams compete on maximal weight loss. Awards are presented weekly based on the top five males and females for the week, and a golden apple award is presented to both the male and female winners based on adherence, motivation, and overall positive mental attitude.

The HEAL program has been continuously improved over the last nine years; attrition is now less than five percent. Applications to the program are submitted electronically. The program director and staff review applications. Approximately 80 individuals are selected to participate in each HEAL session, which is offered twice a year (in January and August).

To enroll in HEAL, participants must meet the program requirement of being at least 15+ pounds overweight and over 18 years old. Participants must also have a valid email address to receive communication containing the online survey link and have internet access to complete the online survey. Finally, participants must agree to provide a pre and post blood sample. The only exclusion criteria for participants was having any conditions that prevented them from participating in vigorous physical activity.

## 2. Materials and Methods

As part of a pilot biomarker study examining the effects of HEAL on AGEs, participants were recruited by the primary investigator during an initial pre-orientation session. The study was explained and a convenience sample approach was utilized to accept the first ten males and first ten females who expressed interest for participation. Twenty primarily European American (19 white, 1 black) participants (i.e., 10 males, 10 females) aged 26 to 71 (m = 48.75 ± 10.26) completed 12 weeks of the HEAL intervention. Baseline anthropometric characteristics are presented in Table 2.

Ethical considerations were put in effect during study completion. All participants completed the consent process in a private area of the wellness center, which was authorized by the Medical University of South Carolina Institutional Review Board. The primary investigator reviewed study procedures with the recruits and explained the risks and benefits of study participation; once this was completed, participants were provided answers to any questions they had. It was further explained to the participants they would need to complete biomarker measurements before and after the HEAL program. All participants who volunteered for the AGE study completed the same HEAL components. Certified personal trainers offered organized exercise sessions focusing on proper biomechanics and functional exercises aimed at improving overall fitness and body composition. Exercise physiologists oversee InBody testing, injury prevention, and education. A registered dietician presents educational sessions using the current research on correct nutrition to achieve healthy body weight and work towards chronic disease prevention. Breathing techniques are taught to aid in stress reduction and sustained success post intervention. Each participant’s specimens were collected within one week prior to beginning the HEAL program and within one week after completion of the HEAL program. The only compensation participants were provided was a free pass for one session with a personal trainer post HEAL intervention and copies of their biomarker results at the end of the study. For the reported pilot study, on each day, biomarker data were collected, participants arrived fasted and met the primary investigator at the wellness center. Participants were escorted to the South Carolina Translational Research Biomedical Nexus for biospecimen data collection before and after their participation in the HEAL program. During each visit, anthropometric measurements and 16mL of blood were collected and analyzed for hemoglobin A-1C and fasting glucose. One tube of blood was transferred to the Medical University Hospital Authority labs and processed for hemoglobin A-1C and fasting glucose. The other tube was prepared for AGE analysis by ELISA assay in the laboratory of Dr. Turner (Cell Biolabs). Each sample was assessed in duplicate. Once blood was collected, participants completed an assessment of total % body fat obtained from dual energy X-ray absorptiometry (DEXA).

## 3. Results

Pre to post changes in BMI, total % body fat, fasting glucose, A1C, and AGEs were examined as primary outcomes using a paired t-test with pre and post intervention data. The results for BMI at pre (32.20 ± 3.91) vs. post (30.18 ± 4.06) HEAL show that participants yielded a significant reduction t(19) = 7.49, *p* = 0.000. In a similar fashion, total % body fat at pre (36.03 ± 5.47) vs. post (32.85 ± 5.82) yielded a significant decrease t(19) = 9.01, *p* = 0.000. Participants’ change in glucose at pre (104.40 ± 22.64) vs. post (98.50 ± 15.61) yielded a nonsignificant reduction t(19) = 1.53, *p* = 0.142. Although change in glucose was not significant, the change in A1C at pre (6.00 ± 1.63) vs. post (5.32 ± 0.59) was significantly lower t(19) = 2.70. *p* = 0.014. The level of change in AGEs measured at pre (72.36 ± 22.32) vs. post (36.32 ± 13.40) was also significant t(19) = 8.72, *p* = 0.000. All participants maintained 100% adherence to the team workouts. Anthropometric and biomarker characteristics were examined across gender with an independent samples t-test; no significant results were found. The t-test for changes in AGEs approached significance t(18) = 1.96. *p* = 0.066. As shown in Table 2, female participants demonstrated a greater reduction in AGEs after completing the intervention.

The dispersion of chronological age groups across the sample was very limited, with 85% of the participants being between 40 and 60 years old. No changes in anthropometric or biomarker data were significantly correlated with chronological age. As shown in Table 3, the only significant correlations were between changes in fasting glucose and AIC, BMI and % body fat and weight, and % body fat with weight. These correlations are expected and are based on similar physiological pathways.

## 4. Discussion

The current study examined the adaptation of a behavioral change intervention program that helps users focus their diet and exercise behaviors to build a healthy empowered active lifestyle (HEAL) to reduce circulatory AGEs. The results from this study provide strong support that the modification of diet and exercise can reduce AGEs in the circulatory system, as well as glucose, A1C, and unwanted body fat.

Changes in AGE levels as a result of the lifestyle intervention were the main outcome of interest, and the average reduction across all participants was 48%. Eighteen participants showed a reduction of AGEs greater than 40%, one participant had a reduction of only 17%, but had a baseline AGE level that was lower than the other participants to start with, and one participant had no change in AGEs. Female participants showed a greater reduction of AGEs than males, but it is unknown if that was due to differences in dietary habits and/or physical activity, or if it was simply because females had higher AGEs levels at baseline. Promisingly, clinical indicators have not yet been established for AGEs, so whether these reductions yield any clinical significance remains to be established. However, it is well established that AGEs, including diet-derived AGEs, can induce genetic mutations [1], protein dysfunction [18], as well as tissue inflammation [19,20] by functioning as a ligand to the receptor of AGE (RAGE) [1,18,19,20]. It is reasonable to postulate that reducing AGE levels in the circulatory system through increased physical activity and diet change will prevent these pathogenic effects, which play a positive role in promoting many chronic disease phenotypes [21,22].

Positive changes in both fasting glucose and hemoglobin A1C were reported from pre to post HEAL examination. Despite the lack of statistical significance for fasting glucose changes from pre (104.40 ± 22.64 mg/dL) to post (98.50 ± 15.61 mg/dL), the reductions found do have clinical utility, as blood glucose charts typically indicate values above 100 mg/dl to indicate impaired glucose handling/prediabetes. A1C is glycated hemoglobin [23]. Like AGEs, changes in hemoglobin A1C were statistically significant, and these reductions corroborate the clinical significance indicated from the reduction of fasting blood glucose levels. In our study, A1C was 6.00 ± 1.63% at pre and 5.32 ± 0.59% at post. As defined by the American Diabetic Association clinical diabetic A1C charts, pre-diabetes is described as A1C values between 5.7% to 6.5%, and values of 6.5% or higher to indicate fully fledged diabetes [24] Nineteen of the participants had values below 5.7% after completion of the HEAL intervention. Similar to the results for AGEs, the inclusion of reported food intake to estimate glycemic index, as well as recording frequency, intensity, time, and type of physical activity, may be further recorded to determine if different exercise modalities have a more positive influence on the improvement of fasting glucose and A1C.

Positive changes were reported for changes in BMI (kg/m^2^) and total % body fat from pre to post HEAL intervention. A statistically significant reduction from 32.20 ± 3.91 at pre-test to 30.18 ± 4.06 at post-test was recorded, there were no changes in height, indicating these changes were from changes in weight. We chose to use BMI instead of weight due to the clinically meaningful predesignated indexes. BMI ranges from 18.5–24.9 are typically classified as normal weight, and ranges from 25–29.9 and 30 or greater represent overweight and obese, respectively. The classification of participants at pre-HEAL intervention included 13 indexed as obese, and 7 indexed as overweight. The classification of participants at post-HEAL intervention included 10 indexed as obese, 8 as overweight, and 2 as normal weight. A criticism of BMI is that it fails to delineate tissue composition of lean and fat mass, as it is computed from an equation that only accounts for height and weight. As a result, total % body fat from a DEXA scan was also included in this study. Total % body fat at pre (36.03 ± 5.47) vs. post (32.85 ± 5.82) yielded a significant decrease, but clinical guidelines for body fat are not well established. Every HEAL trainer incorporates weight bearing exercises into their exercise sessions with a goal for participants to improve their capabilities with these exercises. Due to this inclusion, we purport that changes in BMI are primarily due to changes in % body fat, and not from muscle wasting or negative changes in bone density. Recording frequency, intensity, time, and type of physical activity, as well as including measurements of strength, would further confirm this assumption.

As part of the HEAL program, teams compete on maximal weight loss and awards are presented based on the top five males and females for the week. A potential limitation of the study is that such ranking and reward can influence outcomes which are not sustained long term after the program ends. Noted in the introduction, a follow up study conducted on past participants reported that 70% participate in physical activity greater than 5 times per week and 66% have maintained or continued to lose weight [17]. However, it is clear that the long-term sustainability of the beneficial effects of the HEAL program in reducing AGE levels would need to be assessed in future large cohort studies, particularly those linking AGE reduction to disease phenotypes and outcomes.

The study was devised to be culturally appropriate for implementation in South Carolina, U.S.A., therefore generalizability of the intervention remains to be established. However, in most regions of the world, the adoption of the unhealthy modern dietary patterns that drive food associated AGE formation are occurring at a far greater frequency than healthy dietary patterns, which prevent AGE formation [12,13,25]. This provides rationale for the potential generalizability of interventions that target lifestyle associated AGEs in different populations from different countries. In addition, participants in the HEAL program were encouraged to make general healthy food choices such as eliminating foods from their diet that are highly processed and high in fat and/or sugar, the AGE-laden foods associated with modern dietary patterns worldwide. Similarly, the exercise workouts within the HEAL study were largely left to the individual’s preference and were highly variable, and included aerobic activities (e.g., running, stairs with weights), anaerobic activities (e.g., group circuit training, weight bearing exercises, etc.), and breathing exercises.

The presented data indicate that AGE reduction has the potential to be an informative bio-behavioral biomarker whose reduction may be indicative of beneficial outcomes for multiple chronic diseases in different populations. This could be achieved in future trials targeted at an international audience that may be used to not only confirm the results of this study, but also attempt to stratify the effects of different physical activity and weight loss programs as individual strategies to reduce AGE levels. This can be done through the use of food logs for the estimation of dietary intake of AGEs. Future studies may include reported food intake of AGEs to better determine the independent and collective effects of diet and exercise in diverse international populations. In addition, frequency, intensity, time, and type of physical activity may also be further recorded to determine if different exercise modalities in these populations have more positive influence on the reduction of AGEs.

## 5. Conclusions

The purpose of this pilot study was to examine HEAL to reduce AGEs, BMI, % body fat, fasting glucose, and A1C. The HEAL program was delivered at a local wellness center using a team-based approach that focused on physical activity and dietary education. The HEAL intervention is not in its infancy, and, at the time of the study, HEAL has been completed for at least two sessions a year for over eight years by the wellness center team, and maintained/modified for improvement by the program’s founder and wellness center director (i.e., Janis Newton). Our study confirmed what the study team anticipated based on anecdotal claims from former HEAL participants. The ten-week program yielded some extremely positive biomarker changes, specifically in relation to diabetes pathogenesis. Despite these extremely positive results, it is important to keep in mind that our HEAL participants are typically extremely highly motivated. Participants who sign up for HEAL are well-informed that the program is designed for major lifestyle changes, and out of pocket costs for the program range between $379 and $479 to enroll, depending on whether you are already a member of the wellness center.

Overall, the study demonstrated that participation in HEAL can lead to statistically and clinically significant improvements in circulatory AGE levels, BMI, total % body fat, fasting glucose, and A1C. However, additional studies with longer follow-up and the inclusion of reported food intake to better determine the independent and collective effects of diet and exercise in addition, frequency, intensity, time, and type of physical activity is warranted, and may provide additional valuable information. Additionally, longer follow-up would allow for a correlation assessment between AGE levels and chronic disease outcomes.

## Figures and Tables

**Table 1 ijerph-18-04863-t001:** HEAL Educational Session.

Timeline	Topics and Learning Objectives
Week 1	Pretesting Body Composition Analysis Baseline Fitness Assessment Program introduction, overview and expectations Nutrition Contract, Grocery List and Guidelines
Week 2	Team Building and Comradery Goal Setting Stress Reduction and Improved Sleep How to Navigate the Nutrition Facts Label
Week 3	Sponsor presentation Mentor Testimonial Exercise’s Impact on Brain Health How to Navigate the Nutrition Facts Label
Week 4	Sponsor presentation Mentor Testimonial Insulin Resistance and Sugar Addiction
Week 5	Sponsor presentation Mentor Testimonial Non-Scale Victories Managing Food Cravings
Week 6	Sponsor presentation Mentor Testimonial Goal Monitoring Midway Check-in Accountability
Week 7	Sponsor presentation Mentor Testimonial Health Impacts of Excess Sodium Strategies for holidays and special occasions
Week 8	Sponsor presentation Mentor Testimonial Debunking Exercise and Nutrition Misinformation
Week 9	Sponsor presentation Mentor Testimonial Dietetic Intern Nutrition Presentation
Week 10	Sponsor presentation Mentor Testimonial Strategies for Successful Long-term Behavior Change

**Table 2 ijerph-18-04863-t002:** Anthropometric and Biomarker Characteristic.

**Baseline**	**Weight (lbs)**	**BMI (kg/m^2^)**	**%Body Fat**	**Glucose (mg/dl)**	**A1C (%)**	**AGES (ug/mL)**
Male (10)	235.36 ± 21.38	33.16 ± 2.87	32.26 ± 2.88	111.20 ± 26.70	5.91 ± 0.88	68.51 ± 24.61
Female (10)	180.99 ± 31.89	31.24 ± 4.69	41.71 ± 3.64	97.60 ± 16.32	6.08 ± 2.2	76.22 ± 20.33
Combined (20)	208.18 ± 38.42	32.20 ± 3.91	36.99 ± 5.81	104.40 ± 22.64	6.00 ± 1.63	72.36 ± 22.32
**Post-Intervention**	**Weight (lbs)**	**BMI (kg/m^2^)**	**%Body Fat**	**Glucose (mg/dl)**	**A1C (%)**	**AGES (ug/mL)**
Male (10)	221.21 ± 22.84	31.20 ± 3.30	28.50 ± 3.61	106.60 ± 17.79	5.28 ± 0.19	40.01 ± 17.15
Female (10)	169.16 ± 30.44	29.15 ± 4.64	37.12 ± 4.24	90.40 ± 7.21	5.36 ± 0.84	32.62 ± 7.40
Combined (20)	195.15 ± 37.40	30.18 ± 4.06	32.85 ± 5.82	98.50 ± 15.61	5.32 ± 0.59	36.32 ± 13.40
**Delta Scores**	**Weight (lbs)**	**BMI (kg/m^2^)**	**%Body Fat**	**Glucose (mg/dl)**	**A1C (%)**	**AGES (ug/mL)**
Male (10)	−14.15 ± 7.23	−1.96 ± 0.99	−3.07 ± 1.51	−4.60 ± 18.99	−0.63 ± 0.79	−28.49 ± 19.04
Female (10)	−11.84 ± 8.81	−2.09 ± 1.45	−3.28 ± 1.71	−7.20 ± 16.15	−0.72 ± 1.42	−43.59 ± 15.22
Combined (20)	−12.99 ± 7.93	−2.02 ± 1.21	−3.18 ± 1.57	−5.90 ± 17.21	−0.68 ± 1.12	−36.04 ± 18.48

**Table 3 ijerph-18-04863-t003:** Correlations of Chronological Age and Anthropometric and Biomarker Change Scores.

		Chronological Age	A1C_Delta	AGEs_Delta	BMI_Delta	%BodyFat_Delta	Weight (lbs)_Delta
Chronological Age	Pearson Correlation	1	−0.383	0.183	0.127	0.037	−0.003
	Sig. (2-tailed)		0.096	0.439	0.595	0.877	0.990
	N	20	20	20	20	20	20
A1C_Delta	Pearson Correlation	−0.383	1	0.218	−0.325	−0.371	−0.300
	Sig. (2-tailed)	0.096		0.356	0.162	0.107	0.199
	N	20	20	20	20	20	20
AGEs_Delta	Pearson Correlation	0.183	0.218	1	0.060	−0.362	−0.010
	Sig. (2-tailed)	0.439	0.356		0.803	0.117	0.966
	N	20	20	20	20	20	20
BMI_Delta	Pearson Correlation	0.127	−0.325	0.060	1	0.647 **	0.967 **
	Sig. (2-tailed)	0.595	0.162	0.803		0.002	0.000
	N	20	20	20	20	20	20
%BodyFat_Delta	Pearson Correlation	0.037	0.371	−0.362	0.647 **	1	0.589 **
	Sig. (2-tailed)	0.877	0.107	0.117	0.002		0.006
	N	20	20	20	20	20	20
Weight(lbs)_Delta	Pearson Correlation	−0.003	−0.300	−0.010	0.967 **	0.589 **	1
	Sig. (2-tailed)	0.990	0.199	0.966	0.000	0.006	
	N	20	20	20	20	20	20

**. Correlation is significant at the 0.01 level (2-tailed).

## Data Availability

Data sharing not applicable.

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
