# Peer review of "Examination of the Effectiveness of the Healthy Empowered Active Lifestyles (HEAL) Program on Advanced Glycation End Products"

_ijerph, 2021, doi:10.3390/ijerph18094863_

Round 1
Reviewer 1 Report
This is a pilot study conducted on 20 European American participants to investigate the effectiveness of the HEAL program to reduce circulatory levels of advanced glycation end products (AGES) and asses its relationship to BMI, % Body fat, fasting glucose, and A1C. The findings showed participants had the following average reductions: AGEs 36.04±18.48ug/ml, BMI 2.0±1.2kg/m2, %Body fat 3.18±1.57%, fasting glucose 5.9±17.21mg/dl, and A1C .68±1.11% over twelve weeks of the HEAL intervention; and shows promise and demonstrates the potential for HEAL as a behavioral intervention to improve pre-diabetic and other inflammatory related comorbidities.
I have several concerns and suggestions for the authors that should be addressed to improve the manuscript.
1. The authors reported patients' baseline measures in Table 2 including BMI, %body fat, weight, glucose, HbA1c, and AGES by gender and for the entire study group. Since the patients' age ranges between 27 and 71 years it will be interesting to reported these patients' characteristics by age groups as well.
2. The title of Table 2 is not accurate since the table reports not only anthropometric characteristics.
3. I recommend to summaries and reported the post-measurements along with pre- and post- measures differences and their respective p-values in a table and/or graphs as well.
4. In the abstract the authors stated that " This pilot study investigated the effectiveness of the healthy empowered active lifestyles 14 (HEAL) program to reduce circulatory levels of advanced glycation end products (AGES) and asses its relationship to BMI, % Body fat, fasting glucose, and A1C". However, in the paper I did not find any reported findings related to the relationship between AGES and BMI, % Body fat, fasting glucose, and A1C.
5. The authors acknowledged that the reported results were obtained in a limited dataset. The authors presented only unadjusted quantitative findings. However, do gender and/or age have any effect on the change in BMI, Total % body fat, fasting glucose, A1C, and AGES reductions? Similarly, what is the effect of other important patients' characteristics such as past and/or current history of diseases and conditions, medications, family history of diabetes and/or obesity on outcomes pre- and post- intervention changes?
Author Response
Please see the attached word document with responses to both reviewers. We appreciate your comments to improve the quality of our submission

Reviewer 2 Report
ijerph-1127156-peer-review-v1
- Is the article within the scope of the journal? The paper is within the scope of the journal. It is an interesting topic to examine lifestyle in relation to Advanced Glycation End products but there are some things that could be clearer in the descriptions.
- General points: The article is well structured. My main concern and query with the article is the following:
- Concerns one in the article is the imbalance in description when it comes to weight in relation to other lifestyle factors. Changing your lifestyle is about much more than losing weight. Weight is important as a marker for a healthy lifestyle, but too much focus on weight in relation to changing one's lifestyle in general, including being active and finding ways to change one's behaviour, is more important in the long term. One can get the impression that it is a common weight loss program which I suppose is not the intention?
- Secondly, my biggest concern in this article is this, which also can affect how the results of the study can be interpreted and used. The HEAL program uses ranking and reward. Ranking and reward could be important for people to be motivated to change, but it could also have an effect on how they think about lifestyle in the long term. Obesity problems is often based on the need to include rewards and confirmation in one's life, rewarding oneself with food. When the confirmation disappears, after the program and when participants return to their daily lives, what happens to the motivation in the long run and what signals are sent out from this program? Possibly this is discussed in the various parts of the program e.g. "Managing Cravings", but it needs clarification? Why ranking and rewards are used in the study and what significance it may have needs to be clarified?
- In addition, there is a lack of transparency about the HEAL program as such. There is a general description of the program from previous studies in the introduction, but in the article, there is no connection and clarity about what changes in diet are made in this study. Nor is the degree of activity mentioned or whether the participants have undergone any kind of counselling.
The importance of subject matter and purpose?
- The article addresses and explores an important topic
Abstract
- The abstract describes the different parts and the process in an acceptable way.
Introduction section
- The introduction section gives a short but good introduction to the subject.
Adequacy of the research design; Materials and methods
- A description could make it clearer and more transparent about the program's structure in relation to how the pilot study was conducted.
Result
- The result is briefly presented and would benefit from getting a better and in-depth description. In addition: At the beginning of the result, it is stated which persons participated in the study. That section may fit better in "Materials and Methods". In addition, questions arise about the importance of separating white and black "European American (19 white, 1 black) participants"?
Discussion
- Although information and description of e.g. A1C and BMI are known and self-evident, so these would have needed to be added in the introductory text rather than being in the discussion in this form. Incidentally, there is a clearer anchoring in the discussion around what is stated to be the purpose of the pilot study.
Adherence, to ethical standards?
- The ethical considerations are not mentioned.
Contribution of new knowledge?
- The article contributes to the knowledge but needs some clarifications.
In general
- References are missing in some places e.g. lines 63 and 170.
Author Response

(The authors gave the same response as above.)

Round 2
Reviewer 2 Report
Dear Authors, I have read the new manuscript and the answers from the authors. I still think there is a problem with the wight focus on the study. Also the information about the content in the program could be more explained. However, I also see that the you have tried to be clearer about what have been studied and what needs to be further studied.
Author Response
Dear Authors, I have read the new manuscript and the answers from the authors. I still think there is a problem with the wight focus on the study. Also the information about the content in the program could be more explained. However, I also see that the you have tried to be clearer about what have been studied and what needs to be further studied.
Response: Further details have been added to the text on the content of the program (lines 112-117):
Certified personal trainers offer organized exercise sessions focusing on proper biomechanics and functional exercises aimed at improving overall fitness and body composition. Exercise physiologists oversee InBody testing, injury prevention and education. A registered dietician presents educational sessions using the current research on correct nutrition to achieve healthy body weight and work towards chronic disease prevention. Breathing techniques are taught to aid in stress reduction and sustained success in post intervention
With respect to the esteemed reviewer, I am unable to understand what the issue with the weight focus of the study is. As a result of the previous comments we added more details about the physical activity side of the HEAL program, but a program goal remains weight loss. To rationalize the inclusion of weight loss the following text has been added to the manuscript (Lines 51 to 56):
Meta-analysis studies demonstrate that weight loss in obese and overweight patients with type 2 diabetes is accompanied by reduction in glycated hemoglobin (HbA1c) in a dose dependent manner and is more effective in in populations with poor glycemic control than in well controlled populations PMID: 28417575. In addition, glycation introduces both morphology and functional changes in adipose tissue which are associated with metabolic and vascular alterations as well as insulin resistance PMID: 28417575.
